# Size-dependent steady state saturation limit in biomolecular transport through nuclear membranes

**P. K. Shakhi, M. M. Bijeesh, J. Hareesh, Toby Joseph, P. Nandakumar [ID], Geetha K. Varier [ID] ***

Department of Physics, BITS Pilani K. K. Birla Goa Campus, Goa, India

* geethakvarier@gmail.com

**Data Availability Statement:** All relevant data are within the paper and its Supporting Information files.

## Abstract

The nucleus preserves the genomic DNA of eukaryotic organisms and maintains the integrity of the cell by regulating the transport of molecules across the nuclear membrane. It is hitherto assumed that small molecules having a size below the passive permeability limit are allowed to diffuse freely to the nucleus while the transport of larger molecules is regulated via an active mechanism involving energy. Here we report on the kinetics of nuclear import and export of dextran molecules having a size below the passive permeability limit. The studies carried out using time-lapse confocal fluorescence microscopy show a clear deviation from the passive diffusion model. In particular, it is observed that the steady-state concentration of dextran molecules inside the nucleus is consistently less than the concentration outside, in contradiction to the predictions of the passive diffusion model. Detailed analysis and modeling of the transport show that the nuclear export rates significantly differ from the import rates, and the difference in rates is dependent on the size of the molecules. The nuclear export rates are further confirmed by an independent experimental study where we observe the diffusion of dextran molecules from the nucleus directly. Our experiments and transport model would suggest that the nucleus actively rejects exogenous macromolecules even below the passive permeability limit. This result can have a significant impact on biomedical research, especially in areas related to targeted drug delivery and gene therapy.

## 1. Introduction

The nucleus isolates and preserves the genomic DNA from other parts of eukaryotic cells by a protective sheath of lipid bilayer membrane known as the nuclear envelope [1,2]. The nuclear envelope is punctured by a vast number of pores through which the nucleus communicates with the rest of the cell via molecular transport [3,4]. The nuclear pore complexes (NPC) on the nuclear pores act as a mediator and a regulator of transport of a wide variety of cargo molecules between cytoplasm and nucleus in a highly selective manner [5–9]. Permeability studies and structural studies have shown that the nuclear pore diameter is approximately equal to 9 nm in various cell types [10–19]. It is hitherto assumed that the NPC allows the transport of

**Funding:** The authors gratefully acknowledge the financial support received from the Science and Engineering Research Board (SERB EMR/2016/003687). The authors are thankful for the financial support to the Department of Physics, BITS Pilani Goa campus by the Department of Science and Technology, Govt. of India under the DST-FIST scheme (Ref No. SR/FST/PSI-142/2009; SR/FST/PS-I/2017/21). P.K. Shakhi acknowledges the Council of Scientific and Industrial Research, Govt. of India for a Senior Research Fellowship. Dr. Geetha K. Varier would like to thank the Department of Science and Technology for a research fellowship under the DST-WOS-A scheme (WOS-A/PM-32/2018). The funders had no role in study design, data collection and analysis, decision to publish, or preparation of the manuscript.

**Competing interests:** The authors have declared that no competing interests exist.

small molecules having a size below this limit by way of free diffusion known as passive nuclear transport. Beyond this passive permeability limit, macromolecules like proteins and nucleic acids having sizes up to 26 nm are transported through the NPC via an active mechanism involving energy. This transport that takes place with the help of soluble transport receptors is referred to as active nuclear transport [20]. The intrinsically disordered phenylalanine-glycine-containing polypeptides that occupy the central channel [18,21–23] of the NPC play an important role in regulating active nuclear transport. Multiple pathways are assumed to be involved in the active transport of molecules through the NPC. Most of the proteins that contain a transport signal within it are transported into and out of the nucleus via the karyopherin β mediated transport pathway in a Ran GTP-dependent manner [24–28]. However, there are many proteins that do not carry a transport signal, but are still transported through the nuclear membrane. These proteins are assumed to be transported by piggybacking on a protein carrying the transport signal [29]. Identifying the various transport pathways of nuclear transport adopted by different molecules are still a work in progress. Kinetic studies on nuclear import and export provide the rates of nuclear transport and could throw light on many unanswered questions on the transport process. Of particular importance is the particle size dependence of the nuclear transport rates, and the size-dependent saturation of transport [16]. More importantly, it is pertinent to ask: Do the molecules having a size below the passive permeability limit diffuse freely across the nucleus, or is there any other dynamic process involved even in this limit?

Time-lapse confocal fluorescence microscopy is a powerful technique to study nuclear transport kinetics [13,16]. In this method, dye-labeled cargo molecules are added to the cytoplasm, and variation of the nuclear fluorescence is monitored as a function of time. The fluorescence intensity is directly proportional to the concentrations of the cargo molecules. The rate of change of nuclear fluorescence to cytoplasmic fluorescence helps in determining the nuclear membrane permeability and nuclear pore size. This is done by fitting the time-lapse data to an appropriate diffusion model. In the most commonly used model it is assumed that for molecules having a size less than the permeability limit, the transport follows the simple diffusion equation with the rate constants of transport to the nucleus and back to cytoplasm being equal. The rate of change of concentration inside the nucleus is proportional to the concentration difference between the nucleus and cytoplasm at the instant [13,15,30]. The model suggests that in the steady state, the concentration inside the nucleus is the same as that outside. However, one often finds size-dependent deviation of the experimental data from the steady-state value predicted by the above equation [12,16,31]. To understand the experimental behavior of the passive transport, we conducted studies on the nuclear influx of FITC dye-labeled dextran molecules through the HeLa cell nuclear membrane. Different probe molecules such as proteins, dextrans, and nanoparticles are extensively used in the literature to study the permeability of nuclear membranes[10–12,30,32]. Dextran molecules, used as a model probe in the present study, is an inert and biocompatible polysaccharide that is non-toxic and is widely used for studying the kinetics of nuclear transport [10–12,30]. Dextran molecules are hydrophilic and show little tendency to interact with cellular components and are not degraded in the cell, making them an ideal candidate to study the passive permeability of nuclear membranes [33]. In 1975, Paine *et al.* studied the nuclear envelope permeability of the amphibian oocyte nucleus by measuring the nucleoplasmic flux of microinjected dextran molecules of different molecular weights using autoradiography [10]. In 1983 Peters derived the nuclear pore size of isolated rat liver cells by studying the nuclear influx of fluorescently labeled dextran molecules having different molecular weights using fluorescence microphotolysis [12]. In 1999 Peters and Oliver Keminer studied the passive permeability of a single nuclear pore in the Xenopus oocyte nucleus using an optical single transporter recording

technique and reported that the diffusion channels of NPC are cylindrical in shape and the pore radius lies between 4.1 nm to 6.1 nm [11]. The nuclear transport of biomolecules can be conveniently studied using a permeabilized cell system. The permeabilized cell system allows one to focus on the transport kinetics of a particular transporting molecule, and no other background molecular transport interferes with the experiment. The study using permeabilized cells also helps to find out various transport factors required for the transport of specific molecules and allows us to examine nuclear transport under different conditions. The protocol for the digitonin permeabilization assay was developed by Adam *et al.* in 1990 and since then many groups have successfully applied it to nuclear transport studies [34–36].

In the present work, we study the nuclear transport kinetics of dextran molecules having a molecular weight below the passive permeability limit in a permeabilized HeLa cell system. Detailed modeling of the transport process is carried out by allowing the possibility of active transport taking place even in this size range. Our model fits well with the observed experimental data and explains the deviation of the data from the expected first-order kinetic behavior. Further, considering the possibility of different import and export rates, we introduce a new experimental method to estimate the nuclear export rates by recording the outflow of dextran molecules from the nucleus to the cytoplasm. The results of the nuclear export experiments agree well with our model and the rates observed in the import experiments.

## 2. Materials and methods

In the present work, the HeLa cell line, a cervical cancer cell line, is used for studying the transport of dye-labeled molecules across the NPC. Studies are carried out in the digitonin-permeabilized HeLa cell system [35]. The permeabilization is carried out in such a way that the cell membrane is selectively removed while the nucleus remains intact. The passive nuclear import and export rates are calculated from the analysis of time-lapse images captured by a confocal laser scanning microscope.

### 2.1 Transport molecules

The transport cargo molecules that are chosen for conducting nuclear transport study are fluorescein isothiocyanate (FITC) dye-labeled dextran molecules having different molecular weights of 10 kDa (FD10S), 20 kDa (FD20S), and 40 kDa (FD40S) from Sigma Aldrich Chemicals Pvt. Ltd. FITC- dextran molecules having a degree of dye labeling of 0.003 to 0.02 moles of FITC per mole of glucose and having an excitation maximum at 490 nm and emission maximum at 520 nm is used for the study. The molecules used for the control experiments are Tetramethylrhodamine (TRITC)-labeled dextran molecules having a molecular weight of 70 kDa (Model No. T1162) from Sigma Aldrich Chemicals Pvt. Ltd. TRITC- dextran has a degree of labeling 0.001 mol TRITC per mol Glucose and have an excitation maximum at 550 nm and emission maximum at 580 nm.

### 2.2 Solutions and reagents

The buffer solution for the nuclear transport called transport buffer is prepared from 20 mM HEPES, pH 7.3, 110 mM Potassium Acetate (KAc), 5mM Sodium Acetate (NaAc), 2 mM Magnesium Acetate (MgAc), 0.5 mM Ethylene glycol tetraacetic acid (EGTA), 2 mM Dithiothreitol (DTT), 1 µg/ml of aprotinin, leupeptin, and pepstatin (protease inhibitors). Transport buffer devoid of DTT and protease inhibitors are prepared as 10X stock solution and then sterile filtered and stored at 4˚C temperature. The working 1X concentration of transport buffer that contains DTT and protease inhibitors is prepared freshly on the day of the experiment by diluting it with sterile double distilled water.

In our study, we have used digitonin (RM807, Himedia Laboratories Pvt Ltd.) dissolved in dimethyl sulfoxide as stock. The stock solution is further diluted with the transport buffer for the cell permeabilization assay in the nuclear transport experiment.

Rabbit reticulate lysate (RRL, L4960, Promega) is used as the artificial cytoplasm to fulfill the requirement of cytosolic factors necessary for nuclear transport [36]. The RRL is dialyzed against the transport buffer before using it in nuclear transport studies. The final import mixture for the nuclear transport study contains 50% of RRL. The remaining 50% contains the desired concentration of import cargo molecules prepared in the transport buffer.

## 2.3 Cell culture

We have purchased the HeLa cell line for the current study from the Cell Repository division of the National Centre for Cell Science (NCCS), Pune, India. The cells are cultured [37] in a complete cell culture medium in which 90% Dulbecco's Modified Eagles medium (DMEM) is supplemented with 10% Fetal Bovine Serum (FBS). The cultures are maintained in a humidified incubator at 37°C temperature and a 5% $CO_2$ atmosphere. The cells for the transport studies are grown in confocal dishes (imaging chambers). These imaging chambers are prepared by drilling a 6 mm hole at the bottom of Petri dishes (35x15 mm, 460035, TARSONS). A coverslip is fixed to the bottom of the Petri dish by melting parafilm using a Bunsen burner.

## 2.4 Cell permeabilization assay

For the transport study, we seed ~ 20,000 cells in the imaging chamber around 14 to 20 hrs. before the experiment, and the culture media is replaced with a fresh media 2 hrs. before the experiment. An optimized concentration of 40 μg/ml digitonin in transport buffer is used for cell permeabilization and is freshly prepared on the day of the experiment. Just before the permeabilization, the growth medium is completely removed from the imaging chamber and the cells are washed twice with 1X transport buffer. Washing is followed by complete draining of the transport buffer solution and the digitonin permeabilization is carried out for an optimized time of 5 minutes. After 5 minutes, the permeabilization is stopped by aspirating the digitonin solution and replacing and washing it with a fresh transport buffer.

## 2.5 Confocal imaging of nuclear transport

The time-lapse confocal imaging of the passive nuclear import and export is done using an Olympus confocal microscope (FV 3000, Olympus corporation) and/ or an in-house constructed confocal fluorescence microscope [38]. The import mixture for the transport is freshly prepared before the experiment. The import mixture contains 50% solution volume FITC-dextran molecule in transport buffer and the remaining 50% is RRL. After digitonin permeabilization, an imaging area of the sample that contains well-separated healthy nuclei of enough number (~10–15) is carefully chosen by observing under the white light mode of the confocal microscope and the position of the chamber is locked. Then the objective of the microscope is focused at the equatorial plane of the nucleus and the position of the objective is also locked to avoid focus drift. The transport buffer is carefully removed from the well and the import mixture is added without disturbing the position of the well. The confocal imaging is started in the time-lapse mode of the 488 nm channel just before the addition of the import mixture. Time-lapse confocal fluorescence images of the nucleus are acquired at specific intervals (2 seconds) of time after the addition of the import mixture to the cytoplasm. The total imaging area of the specimen is acquired as 103μm X 103μm with 512 x 512 pixels. The import experiment is conducted for a time of 5 minutes.

The nuclear export assay is conducted after completing the passive import assay procedures. After 5 minutes of import assay, the import reaction is stopped by aspirating the remaining import mixture from the imaging chamber followed by a minimum of three times washing the nucleus with the transport buffer. The washing procedure helps to remove any FITC-dextran molecule present outside the nucleus. The washing step is conducted very gently and is done within a few seconds and the washing time is maintained for all sets of experiments. After washing, a solution mixture of 50% volume of RRL and the remaining 50% volume of complete transport buffer is added. At this stage, the concentration of FITC-dextran molecules will be high inside the nucleus compared to the outside. As a result, the dextran molecules will start exporting from the nucleus to the outside volume. The time-lapse nuclear export imaging is started immediately using a confocal laser scanning microscope.

## 2.6 Control experiment

A control experiment is conducted at the end of each export experiment to verify that the nucleus is intact. The control experiment is conducted using TRITC-labeled dextran molecules of molecular weight of 70 kDa having a size much larger than the passive diffusion limit. The experiment is carried out by adding the import mixture containing 70 kDa dextran to the sample and monitoring the fluorescence increase inside the nucleus. An intact nucleus will not show any increase in fluorescence inside the nucleus with time and will appear dark.

## 2.7 Image analysis

We have used ImageJ software for the analysis of the acquired images. ImageJ is an open-source Java-based image processing software developed at the National Institutes of Health and the Laboratory for Optical and Computational Instrumentation (LOCI, University of Wisconsin). We have analyzed the raw images of nuclear transport by measuring the value of fluorescence inside the nucleus by marking an area covering the entire nucleus excluding the nuclear membrane. The outside fluorescence is measured by measuring fluorescence at multiple areas nearer to the outer portion of the nuclear membrane and taking an average. Normalized nuclear fluorescence is calculated by taking the ratio of average fluorescence inside and outside of the nucleus. Any background noise present is subtracted before plotting and analyzing the normalized nuclear fluorescence. The graph of normalized nuclear fluorescence with time for the nuclear import is plotted using MATLAB software.

## 3. Results

### 3.1 Nuclear import experiments

Nuclear import of FITC labeled dextran molecules (FITC-dextran) having molecular weights of 10 kDa, 20 kDa, and 40 kDa are studied in digitonin permeabilized HeLa cell system. The import mixture containing 50% volume of rabbit reticulocyte lysate (RRL) and 0.5 mg/ml of FITC-dextran is added to the permeabilized cell system after focusing the confocal laser scanning microscope objective at the equatorial plane of the nuclei. Time-lapse confocal fluorescence images of the equatorial plane of the nucleus are acquired at specific intervals after the addition of the import mixture to the cytoplasm. The integrated fluorescence intensity in this volume, being proportional to the number of dextran molecules inside the nucleus, allows us to estimate the rate of nuclear entry of the molecules. The experiment is carried out in a large number of (50 to 70) identically grown HeLa cell nuclei to verify the reproducibility and to determine the mean rate constant of transport and their spread. A control experiment is conducted to verify that the nucleus is intact after the passive import experiment using TRITC-

labeled dextran molecules of molecular weight of 70 kDa having a size much larger than the passive diffusion limit. The experiment is carried out by adding the import mixture containing 70 kDa dextran to the cytoplasm and monitoring the fluorescence increase inside the nucleus. An intact nucleus will not show any increase in fluorescence inside with time.

The size of the dextran molecules studied here is less than the passive permeability limit and is expected to be transported through the nuclear membrane through free diffusion. Fig 1 shows representative confocal images of the nucleus at specific intervals after the addition of the import mixture containing dextran of molecular weight 20 kDa. It can be observed that in the initial frames, the nuclei appear dark in color due to the absence of a significant amount of dye-labeled dextran molecules. As time progresses the nuclei get brighter and brighter until a steady state is reached. To verify that there is no slow diffusion leading to the inside concentration saturating to the value outside, time-lapse confocal imaging for a longer duration is conducted. In Fig 2 we show the confocal images of the nuclei acquired at 5-minute intervals after the addition of transport buffer containing 20 kDa dextran molecules. Sufficient care is taken to ensure that the nucleus is intact even after 40 minutes of exposure. The results of this study show that the fluorescence intensity outside the nucleus is higher than that inside the nucleus even after 40 minutes and that saturation is established well below the expected saturation limit.

In passive nuclear transport, it is expected that the molecules enter the nucleus through the nuclear pore complex by diffusion. To quantify the diffusion parameter, the ratio of the concentration of the dextran inside the nucleus to that outside (normalized nuclear concentration) is calculated. Fig 3 shows a graph of normalized nuclear concentration with time for the nuclear import of FITC-labeled dextran molecules having molecular weights of 10 kDa, 20 kDa, and 40 kDa. From Fig 3, it can be observed that there is a significant difference in the

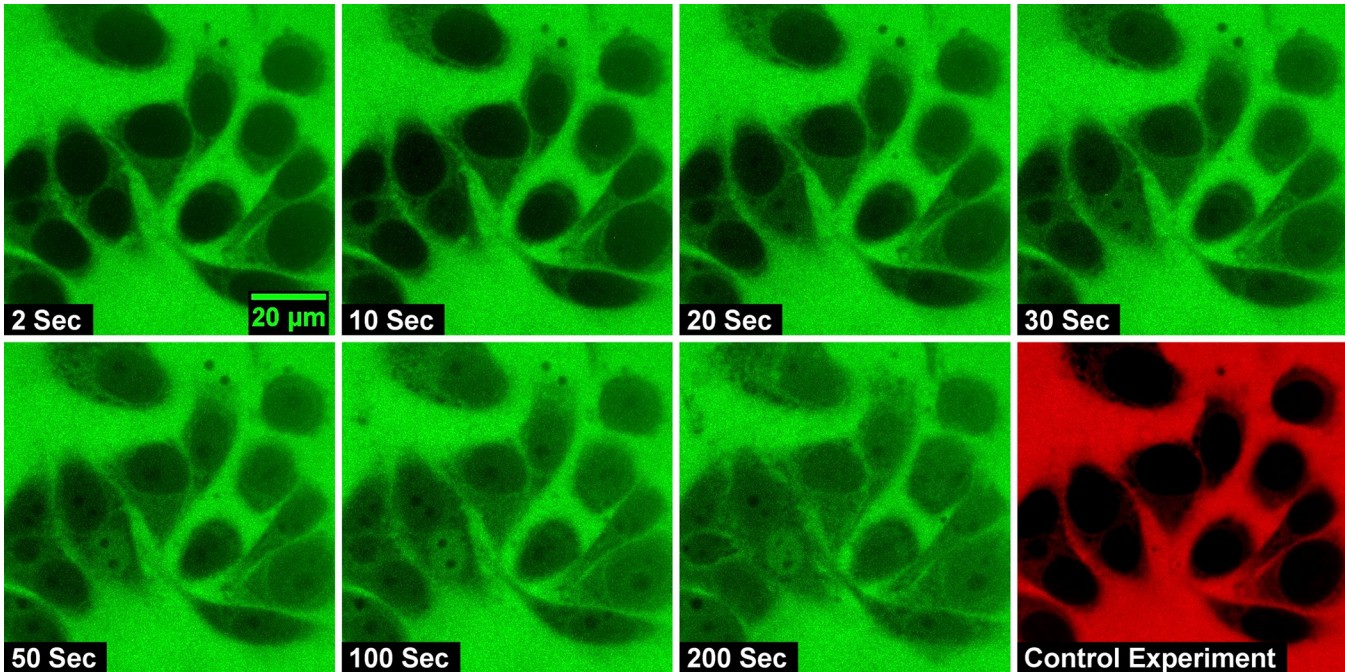

**Fig 1. Time-lapse confocal images of passive nuclear import of 20 kDa FITC-Dextran molecules across HeLa-cell nuclei.** The intensity of the nuclear fluorescence indicates the nuclear entry of the fluorescently tagged dextran molecule. The last frame marked as a control experiment is the image of the nuclear exclusion of 70 kDa TRITC dextran molecules. The dark-colored nuclei indicate that they are intact.

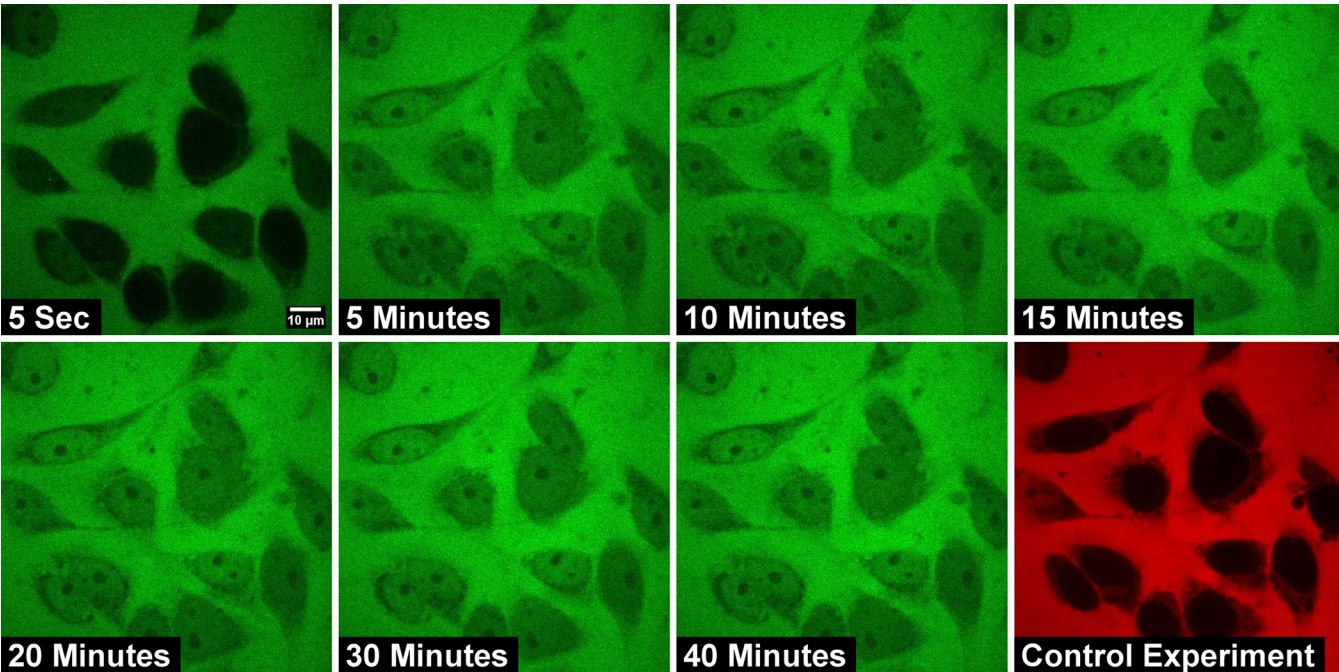

**Fig 2. Time-lapse confocal images of passive nuclear import of 20 kDa FITC-Dextran molecules across HeLa-cell nuclei acquired at different time intervals after the addition of import mixture.** The nuclear import study is carried out for a longer duration, up to 40 minutes to see whether there is any slow diffusion of dextrans to the nucleus. The normalized nuclear intensity after 40 minutes of nuclear transport is 0.5, well below the expected value of 1 expected at saturation.

concentration of dextran molecules between the nucleus and cytoplasm in the steady state. This difference in the steady state is larger for the diffusing particles having a larger size.

The concentration ratio found using time-lapse imaging can be fitted to a first-order kinetics equation of the form,

$$C(t) = C_{max}(1 - e^{-kt}) \qquad (1)$$

Where $C(t)$ is the normalized nuclear concentration of dextran at time $t$, $C_{max}$ is the normalized nuclear concentration at equilibrium, and $k$ is the diffusion rate constant. As per the diffusion process, at equilibrium, the concentration of dextran molecules inside the nucleus should be equal to that in the cytoplasm. This means that the normalized nuclear concentration of dextran at equilibrium ($C_{max}$) should be 1. In contradiction to the above fact, Fig 3 clearly shows that normalized nuclear concentration saturates to a value less than 1. Moreover, the saturation values are decreasing with the increasing size of dextran molecules. This shows that the nuclear transport process needs much more detailed analysis and cannot be attributed to a first-order kinetic equation of the form given in Eq (1). In this context, it is necessary to develop a model of nuclear transport that addresses the stated nonconformity to the first-order kinetic equation.

Let us first consider a situation in which the import and export are happening at the same rates. When the import and export rates are equal, the rate equation can be written as,

$$\frac{dC_i}{dt} = k(C_o - C_i) \qquad (2)$$

Where $C_i$ is the concentration of transport molecules inside the nucleus, $C_o$ is the concentration of transport molecules outside the nucleus in the cytoplasm, $k$ is the rate constant and $t$ is

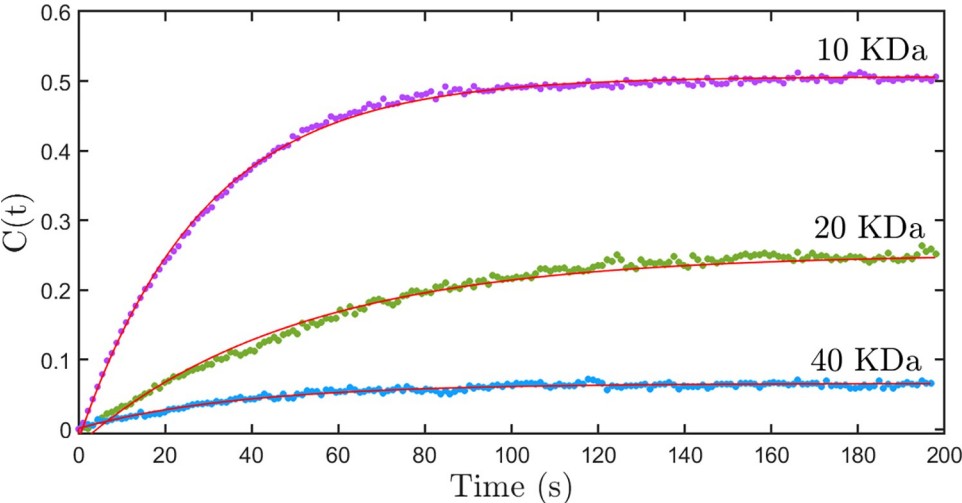

**Fig 3. The graph of normalized nuclear concentration with the time-passive diffusion model.** The graph of normalized nuclear concentration with time for the nuclear import of FITC labeled dextran molecules having a molecular weight of 10 kDa (purple dots), 20 kDa (green dots), and 40 kDa (blue dots). The solid lines are fitted to a first-order kinetics equation of the form $C(t) = C_{max}(1 - e^{-kt})$. The fits show that the $C_{max}$ values differ from the expected value of 1.

time. If we consider the fact that the volume of the nucleus is much smaller compared to the volume outside the nucleus, $C_o$ can be approximately taken as a constant. The solution of the above equation is,

$$C_i(t) = C_o(1 - e^{-kt}) \tag{3}$$

If we divide with $C_o$ we get the normalized concentration, plotted in Fig 2

$$C(t) = \frac{C_i(t)}{C_o} = \left(1 - e^{-kt}\right) \tag{4}$$

This suggests that at a steady state, $C_i = C_o$ and that Eq (4) will be equivalent to Eq (1) if $C_{max}$ is equal to 1. However, the experimental observation is not consistent with this behavior. Eq (2) can be modified by assigning different rates for the import and export process across the nuclear membrane. By doing this, one is assuming that there is some other dynamic process involved in the transport, and is no more attributed to diffusion alone. Consider β as the nuclear import rate and α as the nuclear export rate. Thus, the rate equation, in this case, can be written as,

$$\frac{dC_i}{dt} = \beta C_o - \alpha C_i \tag{5}$$

The solution, assuming that outside concentration is independent of time, is given by,

$$C_i(t) = \frac{\beta}{\alpha} C_0 (1 - e^{-\alpha t}) \tag{6}$$

If we normalize by $C_o$,

$$\frac{C_i(t)}{C_o} = \frac{\beta}{\alpha}(1 - e^{-\alpha t}) = C_{max}(1 - e^{-\alpha t}) \tag{7}$$

Where, $C_{max}$ is given by $\frac{\beta}{\alpha}$. Thus, within this model, the normalized saturation value of fluorescence gives the ratio of the import rate to the export rate. We fit the experimental data given in Fig 3 to Eq (7) to estimate the $C_{max}$ values and to show that the above model predicts the experimental results correctly. The value of $C_{max}$ is consistently found to be less than 1 for all sizes of dextran molecules studied, indicating that the export rate $\alpha$ is larger than the import rate, $\beta$. The larger values of export rates indicate the presence of an active export process which provides an additional contribution to the export. Further, the observation of the monotonically decreasing nature of $C_{max}$ with size indicates that the active transport of molecules out of the nucleus is stronger for larger particles. Though the above modification allows for a consistent interpretation of the value of $C_{max}$, it still does not explain the systematic deviation that one observes in the fit curves (solid line) shown in Fig 3. For example, it may be noticed that in Fig 3 the time-lapse plots for 10 kDa and 20 kDa dextran clearly deviate from the expected first-order behavior (solid line) at 60 seconds and 40 seconds respectively. Such subtle differences are observed repeatedly in most nuclei and for all sizes of particles. These deviations cannot be attributed to the diffusion inside the nucleus since the dextran molecules redistribute uniformly within the nucleus on a time scale much faster than the kinetic time constant.

To explain the systematic deviation in the fit curves, we may consider the possibility that the export rate, $\alpha$, starts from its initial value $\beta$ (corresponding to free diffusion) and increases to its final value $\beta+\delta$ over a time $\frac{1}{\gamma}$. The variation of $\alpha$ with time is modeled as,

$$\alpha = \beta + \delta \tanh(\gamma(t)) \tag{8}$$

Where we have used the tanh function to capture the rise and saturation of the export rate from the initial to the final value.

The curve fitting is performed by MATLAB software. Least square fitting is used to find values for the parameters $\alpha$, $\beta$ for the best possible fit to the experimental data set.

Fig 4 shows the graph of normalized nuclear concentration with time for 10 kDa, 20 kDa, and 40 kDa dye-labeled dextran molecules, along with the fitting curve obtained using the above model. It can be seen that the model fits well with the data and explains the saturation intensity. It also captures the fine deviations of the data from the first-order kinetic model. Fig 5(A), 5(C), and 5(E) show the histograms of the nuclear export rate $\alpha$ obtained from the studies carried out on nuclei for 10 kDa, 20 kDa, and 40 kDa dye-labeled dextran molecules respectively. The corresponding nuclear export rates are 0.045 ± 0.008, 0.018 ± 0.004, and 0.012 ± 0.003 respectively. The histogram of import rate $\beta$ for 10 kDa, 20 kDa, and 40 kDa dye-labeled dextran molecules is shown in Fig 5(B), 5(D), and 5(F) respectively. The values of nuclear import rates obtained are 0.023 ± 0.005, 0.005 ± 0.001, and 0.0010 ± 0.0003. It is observed that the time constant gamma obtained from the least square fits are of the same order as the rate constants of transport, but shows a larger spread.

## 3.2 Nuclear export experiment

The experiment on nuclear import discussed above indicates that steady-state transport rates from the cytoplasm to the nucleus and that from the nucleus to the cytoplasm are different even for molecules having a size less than the passive permeability limit. From the nuclear import experiment, we determined these two rates and found that the export rates are larger than the import rates. The export rate $\alpha$ can also be independently found and verified by

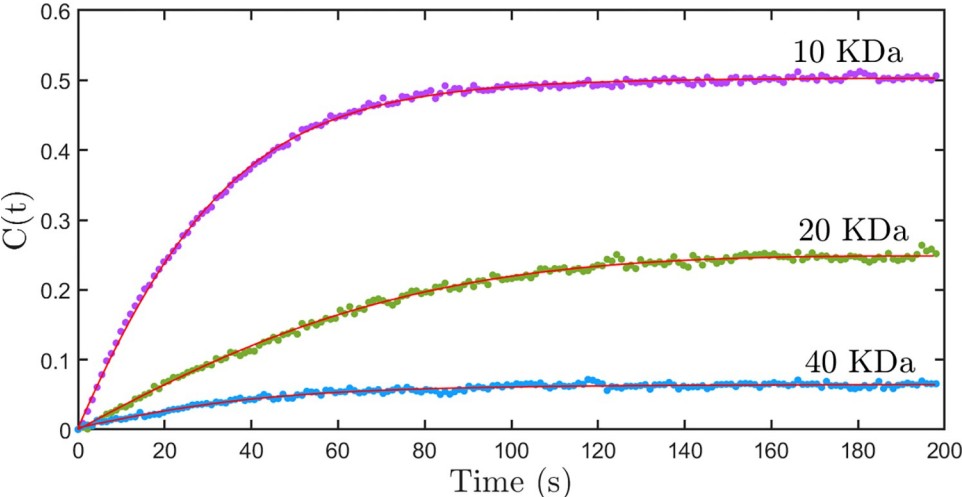

**Fig 4. The graph of normalized nuclear concentration with time–A new theoretical model discussed in the text.**
The graph of normalized nuclear concentration with time for the nuclear import of FITC labeled dextran molecules
having a molecular weight of 10 kDa (purple dots), 20 kDa (green dots), and 40 kDa (blue dots). The solid lines fit the
theoretical model discussed in the text.

performing nuclear export studies. In nuclear export experiments, the dextran molecules are
first transported into the nucleus until a steady state is reached, as discussed in the previous
section. Once the nuclear transport process has reached the equilibrium state, the transport
molecules from the cytoplasm are removed completely. The concentration of transported dye-
labeled dextran molecules is now high inside the nucleus compared to the cytoplasm outside
the nucleus. Due to this gradient, the concentration of the dye-labeled dextran molecules
decreases monotonously and the time-lapse image of the process can be acquired for analysis.

Since the volume outside the nucleus into which the dextran molecule diffuses is much
larger compared to the nuclear volume, one can assume that the outside concentration
remains zero. This implies that the import of dextran back into the nucleus can be neglected.
The rate equation for the nuclear export process can be written as,

$$\frac{dC_i}{dt} = -\alpha C_i \tag{9}$$

Here we assumed that the outside volume of the nucleus is very high compared to the
nuclear volume so that the number of molecules that are imported back to the nucleus is negli-
gible. The solution to Eq (9) is,

$$C_i(t) = C_{io}e^{-\alpha t} \tag{10}$$

Where $C_{io}$ is the maximum dextran concentration in the nucleus corresponding to the
maximum nuclear fluorescence at the start of the export process. The nuclear export rates for
different sizes of dye-labeled dextran molecules can be found and compared with the rates esti-
mated from the nuclear import studies.

Fig 6 shows the time-lapse images of the central cross-section of the nucleus at different
time intervals after the import mixture is removed from outside the nucleus, for the size of 20
kDa dextran molecules. The images show that the concentration of dextran molecules inside
the nucleus decreases with time as expected. The last frame in the image sequence is the con-
trol experiment conducted with TRITC labeled 70 kDa dextran molecules. The frame shows

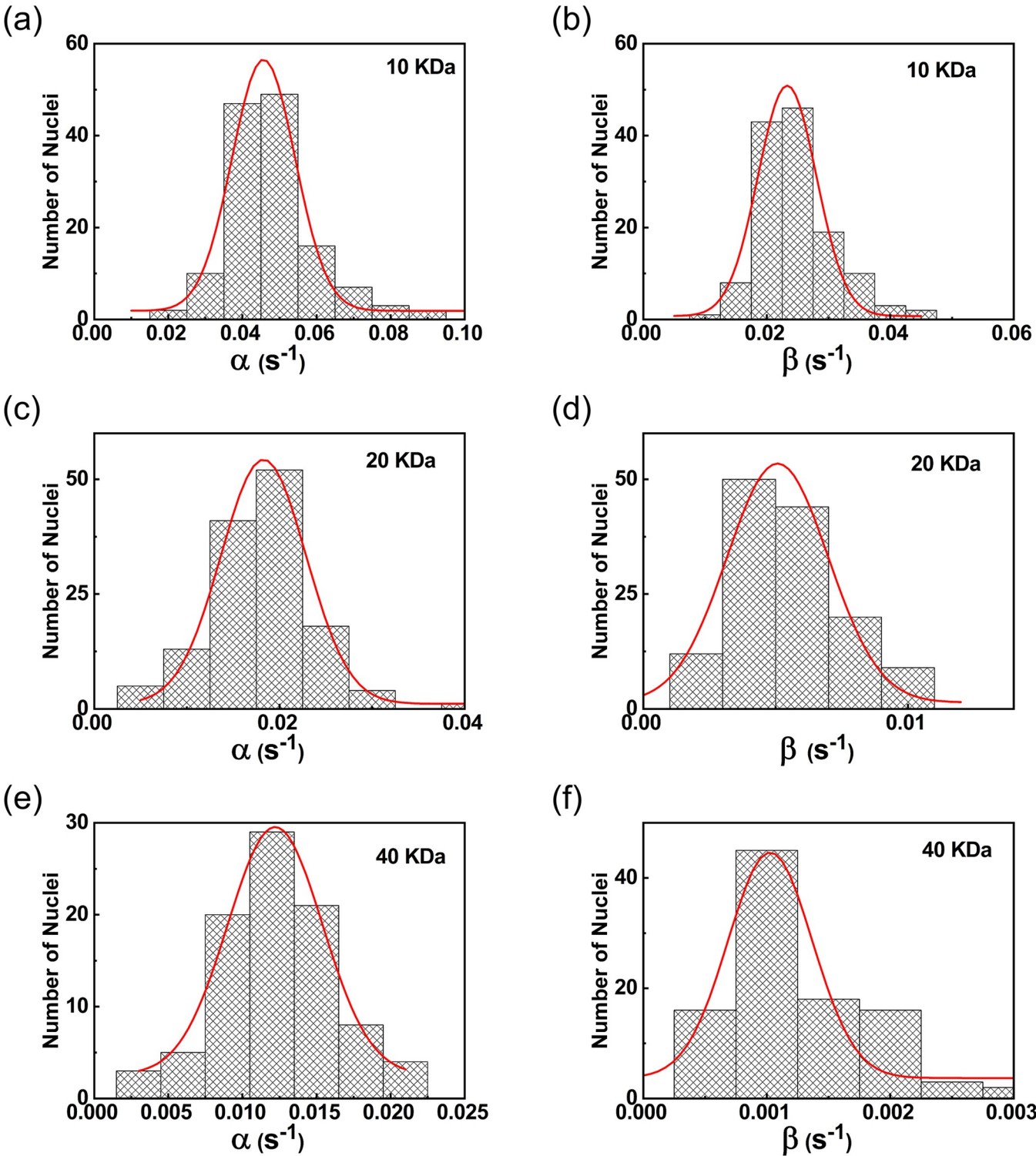

**Fig 5. Histogram of nuclear export rate and import rate deduced from the theoretical fit of nuclear import data. (a), (c), and (e)** show the histogram of the nuclear export rate $\alpha$ and **(b), (d), and (f)** shows the histogram of the nuclear import rate $\beta$ of dextran molecules having a molecular weight of 10 kDa, 20 kDa, and 40 kDa respectively.

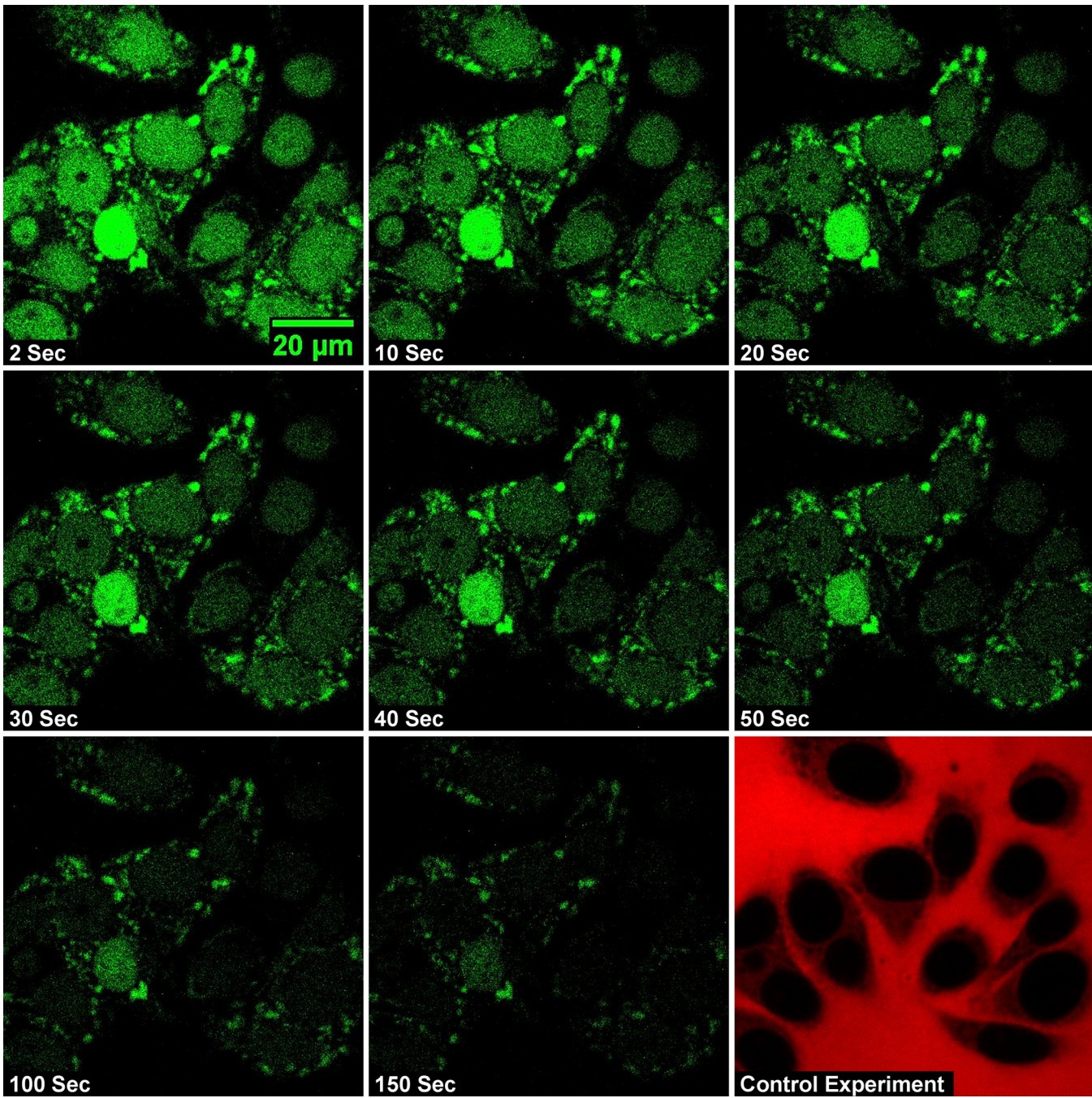

**Fig 6. Time-lapse confocal images of passive nuclear export of 20 kDa FITC-Dextran molecules across HeLa-cell nuclei.** Representative time-lapse confocal images of the central cross-section of the nuclei depicting the export of 20 kDa dextran molecules from the nucleus. The last frame marked as a control experiment is the image of the nuclear exclusion of 70 kDa TRITC dextran molecules. The dark-colored nuclei indicate that they are intact.

that 70 kDa dextran does not enter the nucleus indicating that the nuclear membrane is intact. Fig 7 shows the graph of the integrated nuclear concentration as a function of time for the nuclear export of FITC-labeled dextran molecules having different molecular weights. The solid line is a fit to Eq (10). A histogram of the nuclear export rates obtained from experiments is given in Fig 8. The value of export rates obtained is 0.045 ± 0.005, 0.019 ± 0.002, and

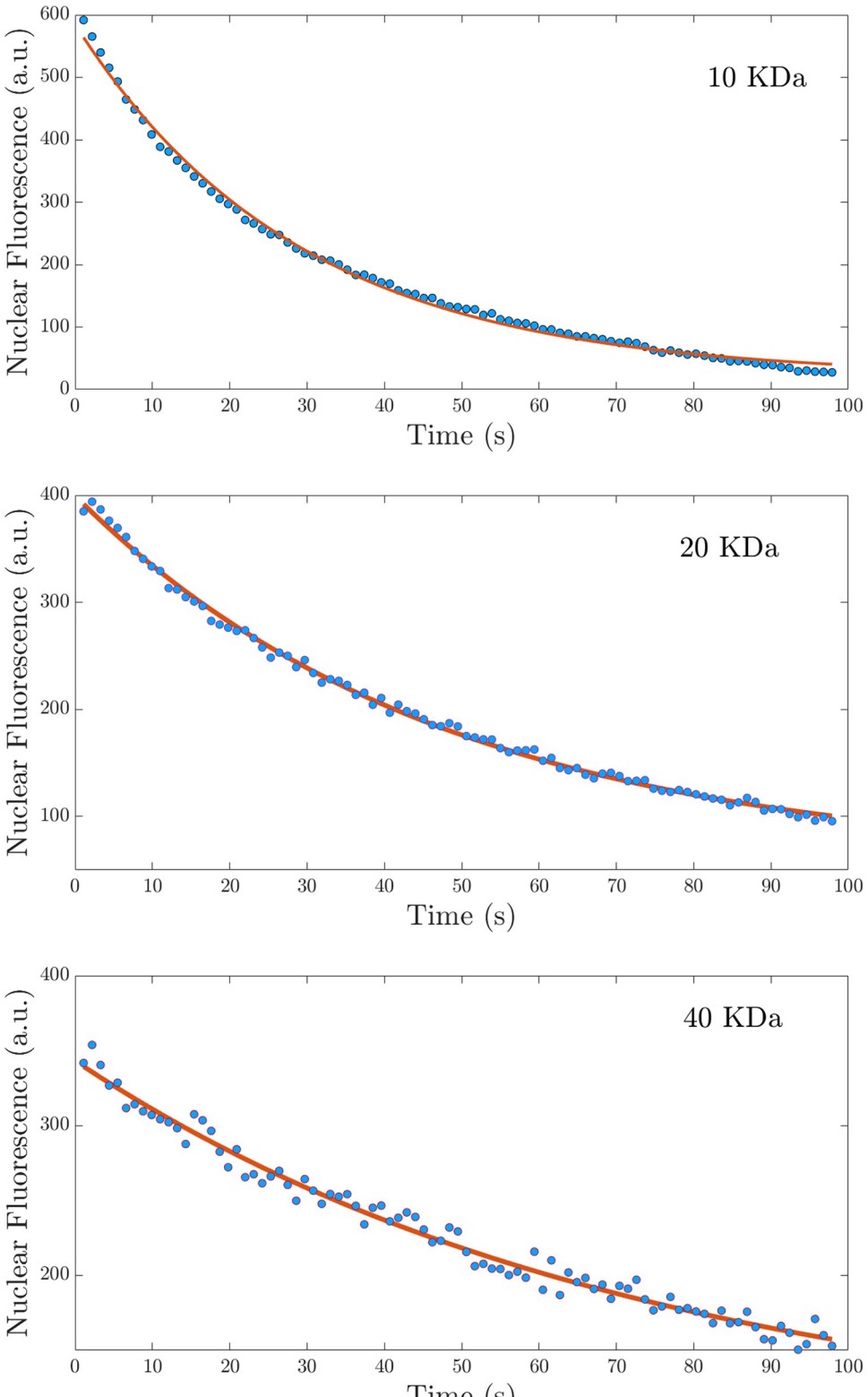

**Fig 7. The graph of nuclear concentration of FITC-Dextran molecules with time for the passive nuclear export studies.** The graph of the concentration of dextran molecules of 10 kDa, 20 kDa, and 40 kDa inside the nucleus as a function of time. The solid lines are an exponential fit to the data.

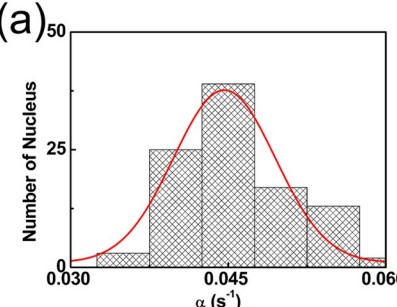 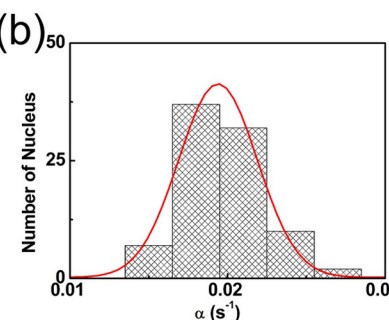 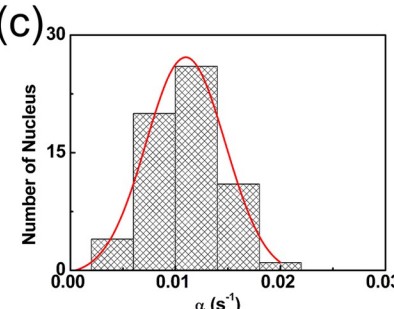

**Fig 8. Histogram of nuclear export rate from the nuclear export data. (a)—(c)** shows the histogram of the nuclear export rate α of FITC labeled dextran molecules having a molecular weight of 10 kDa, 20 kDa, and 40 kDa respectively.

0.011 ± 0.004 for the sizes of 10, 20, and 40 kDa dextran respectively. Table 1 gives the import rate constant $\beta$ and the export rate constant $\alpha$, for the three different-size dextran molecules we have studied. As can be seen from Table 1, these values compare well with the export rates obtained from the nuclear import experiments.

## 4. Discussion

Studies on the kinetics of biomolecular transport across nuclear membranes provide valuable information on the mechanism of transport. Many groups have investigated the transport kinetics of biomolecules across nuclear membranes to infer the passive permeability limit of membranes [10,11,14,15]. These studies carried out using molecules having a size below the passive permeability limit assume that the rate of nuclear import of these molecules is the same as the rate of nuclear export. The permeability of the nuclear membrane and the number and size of the nuclear pores estimated from these studies depend on this assumption of free diffusion. In this work, we critically look at this assumption and examine the kinetics of nuclear import and export of a model system of dextran molecules using time-lapse confocal fluorescence microscopy. Our studies clearly indicate that the nuclear transport of dextran molecules having a size below the passive permeability limit does not strictly follow the hitherto assumed passive diffusion model. In particular, it is observed that the rate of change of concentration of dextran molecules inside the nucleus does not follow the expected first-order kinetic behavior. The steady-state concentration inside the nucleus is found to be less than the concentration outside, in contradiction to the predictions of the free diffusion model. We may also note here that the size-dependent saturation limit reported here is not limited to dextrans, but similar effects have also been observed in the nuclear transport of DNA molecules [39]. Detailed analysis and modeling of the transport data show that the rate constants of nuclear export are greater than the rate constants of nuclear import. The difference between export and import

**Table 1. The nuclear import rate constants and export rate constants of FITC dextran molecules.**

| Dextran size | Stokes radius (nm) | Results from import experiments | | Results from export experiments |
|---|---|---|---|---|
| | | Export rate α (s⁻¹) | Import rate β (s⁻¹) | Export rate α (s⁻¹) |
| **10 kDa** | 2.3 | 0.045 ± 0.008 | 0.023 ± 0.005 | 0.045 ± 0.005 |
| **20 kDa** | 3.3 | 0.018 ± 0.004 | 0.005 ± .001 | 0.019 ± 0.002 |
| **40 kDa** | 4.5 | 0.012 ± 0.003 | 0.0010 ± .0003 | 0.011 ± 0.004 |

The nuclear import rate constant ($\beta$) and export rate constant ($\alpha$) extracted from the experimental studies of the import (columns 3) and export (column 4) of FITC dextran molecules having sizes, 10 kDa, 20 kDa, and 40 kDa. These two independent experiments provide near-identical values for the export rates.

rates is larger for molecules having a larger size. The lower values of the saturation concentration of dextran inside the nucleus observed in time-lapse confocal fluorescence microscopy are due to this size-dependent difference in import and export rates. The difference in influx and outflux rates suggests the possibility that the nucleus actively rejects exogenous molecules in a size-dependent manner. The rate constants of export obtained from independent nuclear export experiments agrees well with those obtained from the import studies.

Active export of macromolecules such as proteins and RNA from the nucleus has been extensively studied in the literature [8,40–44]. Proteins having a nuclear export signal in their structure are recognized by transport receptors such as Karyopherin-β within the nucleus and are exported through the nuclear membrane to the cytoplasm. It is also known that many proteins that do not carry any transport signal can also be actively transported into the nucleus. These proteins are assumed to be transported by piggybacking on transportin-containing proteins [29]. It is plausible that other macromolecules are also exported similarly from the nucleus. An active mechanism of export suggested in our experiments may be explained by a model similar to the piggyback model, but further experimental confirmation of the mechanism of export of these molecules is needed. Identifying the export pathway for this active expulsion of exogenous molecules and their modeling could help to improve the uptake of drugs in cancer treatment, nanoparticles in photothermal therapy, and DNAs in gene therapy.

## Supporting information

**S1 Fig. Time-lapse confocal images of passive nuclear import of 10 kDa FITC-Dextran molecules across HeLa-cell nuclei.**
(TIF)

**S2 Fig. Time-lapse confocal images of passive nuclear import of 40 kDa FITC-Dextran molecules across HeLa-cell nuclei.**
(TIF)

**S3 Fig. Time-lapse confocal images of passive nuclear export of 10 kDa FITC-Dextran molecules across HeLa-cell nuclei.**
(TIF)

**S4 Fig. Time-lapse confocal images of passive nuclear export of 40 kDa FITC-Dextran molecules across HeLa-cell nuclei.**
(TIF)

**S5 Fig. Abstract figure.**
(TIF)

**S1 Raw data.**
(XLSX)

**S2 Raw data.**
(XLSX)

**S3 Raw data.**
(XLSX)

**S4 Raw data.**
(XLSX)

## Acknowledgments

Authors are grateful to Ishan Agarwal, Department of Physics, BITS Pilani K. K. Birla Goa Campus, Goa, India– 403726, for useful discussions. We Acknowledge the Central Sophisticated Instrumentation Facility (CSIF), BITS Pilani K. K. Birla Goa Campus, Goa, India– 403726 for the experimental facilities provided.

## Author Contributions

**Conceptualization:** P. K. Shakhi, P. Nandakumar, Geetha K. Varier.

**Data curation:** P. K. Shakhi, M. M. Bijeesh.

**Formal analysis:** M. M. Bijeesh.

**Funding acquisition:** P. Nandakumar.

**Methodology:** P. K. Shakhi, Geetha K. Varier.

**Software:** M. M. Bijeesh, J. Hareesh, Toby Joseph.

**Supervision:** Toby Joseph, P. Nandakumar, Geetha K. Varier.

**Validation:** Toby Joseph, P. Nandakumar, Geetha K. Varier.

**Writing – original draft:** P. K. Shakhi.

**Writing – review & editing:** M. M. Bijeesh, J. Hareesh, Toby Joseph, P. Nandakumar, Geetha K. Varier.

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
