## [Decision Letter · Decision Letter 0]

21 Nov 2023

PONE-D-23-32002Size-dependent steady state saturation limit in biomolecular transport through nuclear membranesPLOS ONE

Dear Dr. K.Varier,

Thank you for submitting your manuscript to PLOS ONE. After careful consideration, we feel that it has merit but does not fully meet PLOS ONE’s publication criteria as it currently stands. Therefore, we invite you to submit a revised version of the manuscript that addresses the points raised during the review process.

We look forward to receiving your revised manuscript.

Kind regards,

Yash Gupta, Ph.D.

Academic Editor

PLOS ONE

Journal Requirements:

"The authors gratefully acknowledge the financial support received from the Science and Engineering Research Board (SERB EMR/2016/003687), Government of India. P.K. Shakhi acknowledges the Council of Scientific and Industrial Research"

"Dr. P Nandakumar received the financial support  from the Science and Engineering Research Board (SERB EMR/2016/003687), Government of India. URL: https://serb.gov.in/

**Additional Editor Comments:**

Authors need to address comments from both reviewers. Also, a more thorough review of literature is needed to justify use of polysaccharide molecules unlike protein probes in other studies e.g. PMID: 25631821

Reviewers' comments:

Reviewer's Responses to Questions

**Comments to the Author**

1. Is the manuscript technically sound, and do the data support the conclusions?

Reviewer #1: Yes

Reviewer #2: Yes

2. Has the statistical analysis been performed appropriately and rigorously? 

Reviewer #1: Yes

Reviewer #2: Yes

3. Have the authors made all data underlying the findings in their manuscript fully available?

Reviewer #1: Yes

Reviewer #2: Yes

4. Is the manuscript presented in an intelligible fashion and written in standard English?

Reviewer #1: Yes

Reviewer #2: Yes

5. Review Comments to the Author

Reviewer #1: In this paper, the authors conduct research on biomolecular transport through nuclear membranes and explain molecule transport via diffusion doesn't always follow first-order kinetics and there might be active/ specific transport systems. This research is important, well done, and with a few minor revisions, should be published in PLOS ONE

1. Did the authors consider the hydrophobicity of dextran? Do they expect to see similar trends using a hydrophilic/neutral substrate?

2.223-224 – The results of this study show that the fluorescence intensity outside the nucleus is higher than that inside the nucleus even after 40 minutes and that saturation is established well below the expected saturation limit. Authors have to show the 40 mins exposure image data.

3. 410-11 – The difference in influx and outflux rates suggests the possibility that the nucleus actively rejects exogenous molecules in a size-dependent manner- authors measured only dextran, Is it similar to all other molecules?

4. In Figure 4 legend is not clear, authors need to write clearly a, c, e and b, d, f

Reviewer #2: Shakhi et al ‘Size-dependent steady state saturation limit in biomolecular 2 transport through nuclear membranes’

Here are some constructive comments and observations you might consider for the provided results:

1. Your description of the digitonin-permeabilized HeLa cell system and the selective removal of the cell membrane while keeping the nucleus intact is well-explained. It provides a clear understanding of the experimental setup.

2. The selection of FITC-dye labeled dextran molecules with different molecular weights for studying nuclear transport is appropriate. This choice allows for a comprehensive investigation into the size-dependent dynamics of nuclear entry.

3. The detailed step-by-step procedures for cell culture, permeabilization assay, and confocal imaging provide a comprehensive guide for researchers attempting to replicate the experiments. This clarity contributes to the reproducibility of the study.

4. The inclusion of time-lapse confocal images and graphs effectively presents the results. The representative images, along with the normalized nuclear concentration graphs, provide a visual representation of the dynamics of nuclear import.

5. he description of nuclear import experiments is well-structured, with a clear presentation of time-lapse confocal images and corresponding graphs. This visual representation effectively captures the dynamics of nuclear entry for dextran molecules of different sizes.

Here some minor questions:

Provide a clearer justification for choosing specific methods or reagents. Why were certain concentrations or conditions chosen? What is the rationale behind using dyes or molecules?

Overall, your results section effectively combines visual representation, quantitative analysis, and theoretical modeling to provide a comprehensive understanding of the observed nuclear transport dynamics.

6. PLOS authors have the option to publish the peer review history of their article (what does this mean?). If published, this will include your full peer review and any attached files.

Reviewer #1: No

Reviewer #2: **Yes: **Manish Shukla

---

## [Author Response · Author response to Decision Letter 0]

9 Jan 2024

First of all, we would like to thank the referees and the editor for the constructive comments and suggestions given to improve the manuscript. We have answered all the queries raised by the referees and have incorporated changes in the manuscript as suggested by the referees. The comments by the referee (bold fonts) and the reply to the comments are detailed below.

Additional Editor Comments: Authors need to address comments from both reviewers. Also, a more thorough review of literature is needed to justify use of polysaccharide molecules unlike protein probes in other studies e.g. PMID: 25631821

We have incorporated changes in the manuscript by adding relevant literature related to nuclear transport and the use of dextran molecules as a probe. Dextran is a model-neutral molecule that is widely used in nuclear transport studies. We have added the following lines to the manuscript and given the relevant references. (Page no. 3, line no. 89 to line no. 102).

“Different probe molecules such as proteins, dextrans, and nanoparticles are extensively used in the literature to study the permeability of nuclear membranes [10–12,30,32]. Dextran molecules, used as a model probe in the present study, is an inert and biocompatible polysaccharide that is nontoxic and is widely used for studying the kinetics of nuclear transport [10–12,30]. Dextran molecules are hydrophilic and show little tendency to interact with cellular components and are not degraded in the cell, making them an ideal candidate to study the passive permeability of nuclear membranes [33]. In 1975, Paine et al studied the nuclear envelope permeability of the amphibian oocyte nucleus by measuring the nucleoplasmic flux of microinjected dextran molecules of different molecular weights using autoradiography [10]. In 1983 Peters et al derived the nuclear pore size of isolated rat liver cells by studying the nuclear influx of fluorescently labeled dextran molecules having different molecular weights using fluorescence microphotolysis [12]. In 1999 Peters and Oliver Keminer studied the passive permeability of a single nuclear pore in the Xenopus oocyte nucleus using an optical single transporter recording technique and reported that the diffusion channels of NPC are cylindrical in shape and the pore radius lies between 4.1 nm to 6.1 nm [11].”

Reviewer #1: In this paper, the authors conduct research on biomolecular transport through nuclear membranes and explain molecule transport via diffusion doesn't always follow first-order kinetics and there might be active/ specific transport systems. This research is important, well done, and with a few minor revisions, should be published in PLOS ONE. 

We thank the referee for appreciating our work.

1. Did the authors consider the hydrophobicity of dextran? Do they expect to see similar trends using a hydrophilic/neutral substrate? 

Answer: Thank you for this suggestion. We have not studied the effect of the hydrophobicity of molecules on their nuclear transport kinetics in the present work. Dextran is an inert and biocompatible polysaccharide widely used in nuclear transport studies showing little tendency to degrade or bind with the cell proteins. The Dextran molecules used in the present study are hydrophilic in nature having a solubility of 25 mg/ml in water (Reference: https://www.sigmaaldrich.com/deepweb/assets/sigmaaldrich/product/documents/651/359/fd10spis.pdf [1]). We would also like to mention here that we have observed similar behavior in the passive nuclear transport of λ-DNA (unpublished Data) and feel that the deviation from first-order kinetics is not due to hydrophobicity. We agree that the hydrophobicity of transporting molecules could be an important aspect of nuclear transport and would like to look at this in detail separately.

2.223-224 – The results of this study show that the fluorescence intensity outside the nucleus is higher than that inside the nucleus even after 40 minutes and that saturation is established well below the expected saturation limit. Authors have to show the 40 mins exposure image data. 

Answer: Thank you for this suggestion. We have incorporated the figure in the revised manuscript. (Fig 2 in the revised manuscript). The description “In Fig 2 we show the confocal images of the nuclei acquired at 5-minute intervals after the addition of transport buffer containing 20 kDa dextran molecules.” Is added to the text (Page no. 8, line no. 247 to Page no.8, line 249). The following figure caption is added in the revised manuscript (Page no. 8, line no. 259 to line no. 264.) 

“Fig 2. Time-lapse confocal images of passive nuclear import of 20 kDa FITC-Dextran molecules across HeLa-cell nuclei acquired at different time intervals after the addition of import mixture. The nuclear import study is carried out for a longer duration, up to 40 minutes to see whether there is any slow diffusion of dextran to the nucleus. The normalized nuclear intensity after 40 minutes of nuclear transport is 0.5, well below the expected value of 1 expected at saturation.” 

Since Fig 2 is newly added, subsequent figure numbers are modified accordingly.

3. 410-11 – The difference in influx and outflux rates suggests the possibility that the nucleus actively rejects exogenous molecules in a size-dependent manner- authors measured only dextran, Is it similar to all other molecules?

Answer: Dextran is used here as a representative model system of inert biomolecules and is widely used in nuclear transport studies. We have observed similar transport behavior in the passive nuclear transport of linear double-stranded DNA fragments of different molecular weights (unpublished Data). 

 4. In Figure 4 legend is not clear, authors need to write clearly a, c, e and b, d, f 

Answer: Thank you for pointing this out. We have changed the figure captions and legends as pointed out (page no. 11, line no. 356). Because an additional image is added, Fig 4 is renamed as Fig 5 in the revised manuscript.

Reviewer #2: Shakhi et al ‘Size-dependent steady state saturation limit in biomolecular 2 transport through nuclear membranes’ 

Here are some constructive comments and observations you might consider for the provided results:

 1. Your description of the digitonin-permeabilized HeLa cell system and the selective removal of the cell membrane while keeping the nucleus intact is well-explained. It provides a clear understanding of the experimental setup. 

2. The selection of FITC-dye labeled dextran molecules with different molecular weights for studying nuclear transport is appropriate. This choice allows for a comprehensive investigation into the size-dependent dynamics of nuclear entry.

 3. The detailed step-by-step procedures for cell culture, permeabilization assay, and confocal imaging provide a comprehensive guide for researchers attempting to replicate the experiments. This clarity contributes to the reproducibility of the study. 

4. The inclusion of time-lapse confocal images and graphs effectively presents the results. The representative images, along with the normalized nuclear concentration graphs, provide a visual representation of the dynamics of nuclear import. 

5. The description of nuclear import experiments is well-structured, with a clear presentation of time-lapse confocal images and corresponding graphs. This visual representation effectively captures the dynamics of nuclear entry for dextran molecules of different sizes. 

Here some minor questions:

Provide a clearer justification for choosing specific methods or reagents. Why were certain concentrations or conditions chosen? What is the rationale behind using dyes or molecules? Overall, your results section effectively combines visual representation, quantitative analysis, and theoretical modeling to provide a comprehensive understanding of the observed nuclear transport dynamics.

Answer: Thank you very much for the positive comments and suggestions. We have incorporated the suggestions in the revised manuscript by adding the following lines to the manuscript and giving the relevant references. (Page no. 3, line no. 103 to line no. 105; Page no. 4, line no. 106 to line no. 112).

“The nuclear transport of biomolecules can be conveniently studied using a permeabilized cell system. The permeabilized cell system allows one to focus on the transport kinetics of a particular transporting molecule, and no other background molecular transport interferes with the experiment. The study using permeabilized cells also helps to find out various transport factors required for the transport of specific molecules and allows us to examine nuclear transport under different conditions. The protocol for the digitonin permeabilization assay was developed by Adam et al in 1990 and since then many groups have successfully applied it to nuclear transport studies [34–36]. In the present work, we study the nuclear transport kinetics of dextran molecules having a molecular weight below the passive permeability limit in a permeabilized HeLa cell system”.

References

1. Zarrintaj P, Saeb MR, Jafari SH, Mozafari M. Chapter 18 - Application of compatibilized polymer blends in biomedical fields. In: A.r. A, Thomas S, editors. Compatibilization of Polymer Blends. Elsevier; 2020. pp. 511–537. doi:10.1016/B978-0-12-816006-0.00018-9

---

## [Editor Report · Decision Letter 1]

12 Jan 2024

Size-dependent steady state saturation limit in biomolecular transport through nuclear membranes

PONE-D-23-32002R1

Dear Dr. K. Varier,

We’re pleased to inform you that your manuscript has been judged scientifically suitable for publication and will be formally accepted for publication once it meets all outstanding technical requirements.

Kind regards,

Yash Gupta, Ph.D.

Academic Editor

PLOS ONE

---

## [Editor Report · Acceptance letter]

2 Apr 2024

PONE-D-23-32002R1 

PLOS ONE

Dear Dr. K. Varier, 

I'm pleased to inform you that your manuscript has been deemed suitable for publication in PLOS ONE. Congratulations! Your manuscript is now being handed over to our production team.

Kind regards, 

on behalf of

Dr. Yash Gupta 

Academic Editor

PLOS ONE